# A Diagnosis of Denial: How Mental Health Classification Systems Have Struggled to Recognise Family Violence as a Serious Risk Factor in the Development of Mental Health Issues for Infants, Children, Adolescents and Adults

**DOI:** 10.3390/brainsci7100133

**Published:** 2017-10-17

**Authors:** Wendy Bunston, Candice Franich-Ray, Sara Tatlow

**Affiliations:** 1Wb Training and Consultancy, P.O. Box 750, Moonee Ponds 3039, Victoria, Australia; 2La Trobe University, Bundoora 3086, Victoria, Australia; 3Mental Health, The Royal Children’s Hospital, 50 Flemington Road, Parkville 3052, Victoria, Australia; candice.franichray@rch.org.au (C.F.-R.); sara.tatlow@rch.org.au (S.T.); 4The Murdoch Childrens Research Institute, Flemington Road, Parkville 3052, Victoria, Australia; 5Department of Paediatrics, The University of Melbourne; Level 2 West Building, The Royal Children’s Hospital, 50 Flemington Street, Parkville 3052, Victoria, Australia

**Keywords:** infants, children, adolescents, family violence, mental health treatment, diagnostic classification of disorders, DSM-5, ICD-10, DC:0-5, CAMHS

## Abstract

Child and adolescent mental health services (CAMHS) routinely overlook assessing for, and providing treatment to, infants and children living with family violence, despite family violence being declared endemic across the globe. As contemporary neuro-developmental research recognises the harm of being exposed to early relational trauma, key international diagnostic texts such as the DSM-5 and ICD-10 struggle to acknowledge or appreciate the relational complexities inherent in addressing family violence and its impacts during childhood. These key texts directly influence thinking, funding and research imperatives in adult services as well as CAMHS, however, they rarely reference family violence. Their emphasis is to pathologise conditions over exploring causality which may be attributable to relational violence. Consequently, CAMHS can miss important indicators of family violence, misdiagnose disorders and unwittingly, not address unacceptable risks in the child’s caregiving environment. Notwithstanding urgent safety concerns, ongoing exposure to family violence significantly heightens the development of mental illness amongst children. CAMHS providers cannot and should not rely on current diagnostic manuals alone. They need to act now to see family violence as a significant and important risk factor to mental health and to treat its impacts on children before these develop into enduring neurological difficulties.

## 1. Introduction

The prevalence of violence within families is considered to be at endemic levels across the world [1,2,3]. Relational trauma and exposure to toxic stress—in utero and perinatally—has been shown to have enduring and detrimental impacts across development within the early years, childhood, and beyond, significantly increasing the development of mental health disorders [4]. This invited paper provides a commentary on the misalignment between current knowledge regarding early brain development and the application of this knowledge in key mental health diagnostic texts in determining, or failing to determine, responses to children impacted by familial violence. It is standard practice across western child and adolescent mental health services to use the criterion of two significant classification and diagnostic manuals in their assessment and treatment plans for children and young people. These two texts are the DSM-5 (Diagnostic and Statistical Manual of Mental Disorders) produced by the American Psychiatric Association [5], and the ICD-10 (International Statistical Classification of Diseases and Related Health Problems) produced by the World Health Organisation [6].

A review was undertaken systematically to ascertain how family violence is referred to within the DSM-5 and the ICD-10. The outcome of these reviews is provided. An additional review was also undertaken of a third important classification manual used when working within infant mental health, the DC:0-5 (Diagnostic Classification of Mental Health and Developmental Disorders of Infancy and Early Childhood) [7]. This small, lesser known, but significant classification manual was developed by workers in the field of infant mental health to help Child and Adolescent Mental Health Services (CAMHS) think about the early experience of infants and young children. The focus of this third text is on classifying behaviours consistent with diagnoses in the DSM-5 and ICD-10 but where early onset is indicated [7]. The results of the review of these three diagnostic classification manuals are provided and the omission of any clear references to family violence within the reviews discussed. The paper concludes that CAMHS, adult mental health services and indeed the infants, children and families impacted by family violence, cannot afford to wait for traditional mental health classification manuals to adequately capture and report on the complexities and risk factors associated with family violence. CAMHS needs to recognise and respond to the scientific research and prevalence evidence now to offer appropriate and timely treatment responses for children. Furthermore, adult mental health services need to name and assess for family violence, remaining cognisant of the fact that the children of their patients who experience family violence are likely to also be victims of that violence.

### 1.1. Prevalence, Causes and Impacts of Family Violence

Whilst debate exists around defining family violence, the distinction is that “in contrast to other forms of violence … relationships usually exist between family violence victims and perpetrators prior to, during, and after violent incidents” [8] (p. 599). Despite differences in terminology including “domestic violence”, “intimate partner violence” (IPV), “wife battering”, “violence against women and children” [8,9,10,11,12,13] there is consensus across the globe that violence within intimate relationships has reached endemic proportions [2,3,10,14,15,16,17,18,19]. The research to date overwhelmingly identifies violence, and/or homicide, which occurs within intimate partner relationships as gendered, with women most at risk of harm by their male partners [3,20]. According to the UN Women’s Council, “One in three women will experience some form of physical and/or sexual violence in her lifetime” [20] (p. 16). This does not mean that men themselves do not experience violence within heterosexual interpersonal relationships [15,21,22,23], nor that violence is not experienced within same sex relationships [24,25,26].

Explanations for what causes violence within families remains complex, with societal attitudes, gender and cultural inequalities, economic pressures and intergenerational transmission of interpersonal violence often cited as significant contributors [10,13,27,28]. Homelessness, alcohol and substance abuse issues, as well as mental health difficulties are also identified as serious risk factors associated with the prevalence of family violence [10,27,29,30,31,32,33,34]. The costs of family violence to society is monumental. Notwithstanding the social and health implications for the individual, the family and the community, the economic costs to societies across the globe are almost incalculable. Various countries across the world have estimated that economic costs of violence against women and their children to be in the billions [11,35,36,37,38,39]. The use of violence within any intimate relationships is increasingly condemned in most societies today [3]. The impact of family violence on infants, children and adolescents is receiving increasing attention and the inherent detrimental implications for their health and wellbeing over time is becoming better understood [40,41,42,43,44,45,46]. Whether infants, children and adolescents reside in a family where their parent is heterosexual, or same-sex, or caregivers are extended family members or otherwise, living with family violence exposes them to an unacceptable risk of harm [3,29,35,47,48].

### 1.2. Infants, Children’s and Adolescent’s Exposure to Violence

Measuring the number of infants, children and adolescent’s present and/or as the direct victim of family violence has been difficult to ascertain. This is often due to a failure to collect reliable data specific to children [44]. Nevertheless, Lieberman, Chu, Van Horn and Harris [43] contend that empirical evidence demonstrates that children under five are more likely than older children to be exposed to trauma, including that which is caused by domestic violence. This is because younger children, and infants in particular, are more likely to be in the immediate care of, or in close proximity to their mothers during violent episodes perpetrated by partners [49]. The younger the child the less capacity they have to protect themselves, flee the violence or be in other environments when violence occurs, such as school, after school care etc. Research has indicated that some women become the victims of violence once they fall pregnant, with the father of the baby more commonly, though not exclusively, identified as the perpetrator [50,51]. Violence during pregnancy increases the risk of infant mortality, premature births and low birth weights [52,53,54,55,56]. In teenage pregnancy, the prevalence of partner violence and increased risks to mother and infant are particularly high [53,57,58,59]. It is estimated that 1 in 4 children will be exposed to family violence in their lifetime [29,47,60]. Australian women reported children being present up to 31% of the time during episodes of violence by their partners [61]. A survey conducted across the European Union reported that 73% of women who had experienced partner violence believed their children were aware of the violence [10].

The World Health Organisation (WHO) states that violence is preventable and has identified six clear strategies to effect this. The first two of these imperatives directly concern the early years and involve “developing safe, stable and nurturing relationships between children and their parents and caregivers” and “developing life skills in children and adolescents” [3] (p.viii). The United States National Intimate Partner and Sexual Violence Survey agrees. They state that the key to prevention may be through developing “Strategies that support the development of safe, stable, nurturing relationships and environments for parents or caregivers and their children” [15] (p.5). The sheer volume of infants, children and adolescents, who with their mothers and primary caregivers are exposed to family violence, warrants a comprehensive treatment response by CAMHS.

### 1.3. Research on How Family Violence and Relational Trauma Impacts Infant and Child Development

Family violence creates relational trauma for all family members, as it disrupts and disturbs all relationships within the family, not just the relationship between warring parents/adults. The quality and continuity of early caregiving experiences has been proven to be crucial to the development of social, emotional, physical and mental health in children [62,63,64,65,66,67,68,69]. Early infant/parent relationship research is often commonly understood in terms of the attachment the infant develops with their primary caregiver, and how this then directly impacts the infant’s neurological, physiological, psychological and emotional development [70,71,72,73,74,75]. Schwerdtfeger and Goff [76] found that where expectant mothers had a history of interpersonal trauma they “reported significantly higher trauma symptoms and lower prenatal attachment than those who reported no history of interpersonal trauma” (p. 46). Research into exposure to domestic violence during pregnancy was also found to negatively impact the way mothers’ see their infant [77,78]. Furthermore, ongoing exposure to family violence was associated with the child developing an insecure attachment with their mother by the age of four [79,80,81]. Exposure to family violence creates relational ruptures which can interrupt the healthy formation of safe and reliable bonds between children and their parent/s [78,79]. This impacts subsequent social relationships, and produces adult attachment behaviours including an oversensitivity to rejection, avoidant and ambivalent patterns of relating and increased risk of replicating violence in intimate relationships [82,83,84]. Attachment theory pioneer John Bowlby considered family violence a “disorder of attachment” [85]. He believed enormous psychological damage was done to the child and the family system, and was puzzled as to why “family violence as a causal factor in psychiatry should have been so neglected” (p. 9).

Where the infant or very young child is traumatised by the violence of one or both parents and/or caregivers there is little to nowhere for the infant or young child to seek safety and protection; as they are dependent on the very caregiving system which is generating the trauma [82,86,87,88,89]. Early exposure to interpersonal trauma such as family violence impacts the emerging subjectivity of the infant and impinges negatively on their developing mental states, capacity for affect regulation and has implications for their later forming executive functioning [40,90,91,92,93,94,95]. Levendosky, Bogat and Martinez-Torteya [42] found that “children are affected by the IPV they witness and often show a traumatic response. The expression of traumatic symptoms is likely to increases as children age; this is consistent with the trajectory of other anxiety disorders and internalising disorders generally” (p. 195). Thus, exposure to violence from birth impacts the developing regulatory capacities of the infant, with increasing socioemotional difficulties being displayed by 12 months [91]. Children up to the age of 8 years have been found to have lower cognitive scores than their peers who were not exposed to family violence [40]. In particular, internalising and externalising difficulties become more apparent for children as they approach school age and beyond, evidenced through anxious, avoidant and/or disruptive behaviours [42,92]. Children and adolescents exposed to ongoing family violence are increasingly recognised within the literature as manifesting symptoms consistent with Post Traumatic Stress Disorder (PTSD) and other psychiatric disorders [46,96,97,98,99,100].

### 1.4. The Science of Brain Development

Research into the impacts of trauma on the developing brain has been significant [4,63,64,74,86,101,102,103]. The developing infant brain is decidedly ‘experience dependent’ and shaped by the caregiving environment [4,62,104]. “Infants and young children exposed to chronic stress or traumas may have increased levels of the stress hormones cortisol, epinephrine and norepinephrine; chronic high levels of these hormones can have negative effects on emotional regulation, cognitive development, and brain development” [4] (p. 386). Further, crucial to healthy infant development is the regulatory role played by the hypothalamic-pituitary-adrenal (HPA) axis and the associated neuroendocrine responses to stress [105]. Disturbances in the functioning of the HPA axis have been found to have particular implications for the development of neuropsychiatric conditions [106]. In addition, ongoing and significant early relational trauma can impact synaptic growth, reduce hippocampal, cerebellar, and corpus callosum volume, as well as risk damaging limbic regions and the prefrontal cortex; deficits consistent with paediatric posttraumatic stress disorder [107].

Direct research on how family violence impacts the infant brain has been inferred from research into the impact of trauma and stress on the developing brain generally [4,95,103]. However, research using neuroimaging with mothers who have been diagnosed with interpersonal violence related PTSD has found their capacity to read their infants social cues is limited [108]. It is the caregiving relationship that operates as the primary organiser in how the infant brain develops. Where this is impacted by the mother’s own trauma [73,86,109] and further where the infant is a witness “it is likely that exposure to IPV can impact a very young child’s neurological development in detrimental ways that in turn can impact other domains of development” [4] (p. 834). Neuroimaging of infants during ordinary sleep found that even exposure to angry verbal conflict alone elicited heightened responses in the brain regions concerned with emotional regulation and processing, implying that even moderate stress can impact infant brain functioning [110]. There is little doubt that acute, ongoing stress affects neural circuity, and no more so than during the early years [74,95,101,111,112,113]. In light of the mounting evidence regarding the impacts of family violence on infants, children and adolescents, how mental health services assess and treat these impacts is a key concern.

## 2. Combined Methods and Results

### 2.1. Referencing Violence within DSM-5

The DSM-5 is nearly 1000 pages long and is separated into three different sections and an appendix. The principal section is Section II, which is 696 pages and lists all mental disorder classifications. This section includes 21 overarching ‘diagnostic categories’ and an additional section of conditions that may be clinically relevant but are not mental disorders. It explains the criteria for these categories and supplies their corresponding ICD-9 and ICD-10 codes.

Section II: Diagnostic Criteria and Codes was reviewed systematically for any mention of the terms “family violence”, “domestic violence”, “interpersonal violence” and “partner violence”. Further, any specific reference of the word “violence” was examined and if there was any indication that this could refer to being a victim or perpetrator of family violence this was included (note, however, that terms such as “sexual violence” and “physical violence” in the context of child abuse were excluded unless it was specific to a family violence context). “Community violence” also emerged, however, this term was excluded from the review as it was felt to reflect violence in the wider community rather than within the family. Additional, potentially related terms such as “adverse events” and “trauma” were also reviewed to establish any association with family violence. During the initial review process, mention of “spouse/spousal beating” also emerged as a descriptor which could be appropriately defined as linking with the term “family violence”. This term was then added to the search. The results can be found in Table 1 below. Terms relating to “Abuse” (including “childhood abuse”, “physical abuse”, “sexual abuse”, “neglect” and “maltreatment”) were also systematically reviewed. Although they are also significant stressors impacting mental health they were not included in the final table of findings or the discussion presented below as a connection to family violence was not specified.

Section II concludes with a final ’catch-all’ category titled “Other Conditions That May Be a Focus of Clinical Attention” [5] and it is noted in the text that these conditions are not mental disorders but “may affect the patient’s care” (For the sake of continuity, the word “patient” will be used forthwith over other possible descriptors such as “consumer” or “client”.) (p. 715) and can be coded “if it is a reason for the current visit or helps to explain the need for a test, procedure, or treatment” (p. 715). It is here that there is more attention to family violence related issues, specifically with regards to violence occurring within intimate (adult) relationships. Spouse or Partner Violence either Physical or Sexual as well as Spouse or Partner Neglect and Spouse or Partner Abuse Psychological are included in this section and signify whether the patient is or has been a victim of intimate partner violence, or, has been a perpetrator and whether it is either confirmed or suspected. In Spouse or Partner Violence, a description is given which specifies events that have occurred over the past year [5], however, the codes attached to this description simply states “personal history (past history) of spouse or partner violence...” (p. 720) making it unclear to the reader if this acknowledges any possible long term impacts on the individual. There is also no mention, where violence has occurred over the past year, of any children of the patient being potentially impacted by this very same violence.

The term “family violence” was not mentioned at all in the mental disorders categories in Section II of the DMS-5. The term “partner violence” was, however, included in the diagnostic criteria for three disorders in the Sexual Dysfunction category and in the Other Conditions That May Be a Focus of Clinical Attention category. “Domestic violence”, “interpersonal violence”, “violence” and “spouse/spousal beating” were mentioned an additional eleven times with two being that of a possible perpetrator. These were in the following disorders: PTSD; Acute Stress Disorder; Depersonalization/Derealisation Disorder; Somatic Symptoms and Related Disorders; Conduct Disorder; and Antisocial Personality Disorder.

As there are 152 mental disorders listed in the DSM-5 (excluding disorders that are ‘Not Otherwise specified’ or ‘Other specified/unspecified’), [5,114] there are many disorders where family violence could readily have been included yet was not. In particular, it was surprising that there were no links made to interpersonal or family violence as a risk factor in the following disorders given that the research suggests otherwise: Schizophrenia Spectrum and Other Psychotic Disorders [115,116,117], Depressive Disorders [118,119,120,121], Anxiety Disorders [118,121], Substance-Related and Addictive Disorders [34,122,123,124,125], Personality Disorders [126,127]. However, this absence of referring to family or interpersonal violence may also be attributable to the use of terminology within the DSM-5 itself. For example, within Anxiety Disorders imprecise and broad terms were used. The term “life stress” was referred to in Separation Anxiety Disorder; “interpersonal stressors” were referred to in Panic Disorder; the term “negative events in childhood” for Agoraphobia and “stressful life situations” was mentioned in Rumination Disorder. That there is no specific mention of family, domestic or interpersonal violence in childhood within Anxiety Disorders, however, is perplexing given the avalanche of research identifying anxiety in children exposed to family violence [4,42,43,45,46,94,95,96,99,128].

When the terms “domestic violence”, “interpersonal violence”, “partner violence”, “violence”, “adverse” and “trauma” were mentioned they were usually referred to in the Risk and Prognostic Factors Section of each disorder. This section is well suited to including family violence. When completing the review of the text it was noted that the Risk and Prognostic Factors Section was not the same for different diagnostic categories with assorted terminology used, distinct focuses and varying depth of description. It is possible that this reflects the different working groups that developed the current DSM-5 [5]. Where “partner violence” was mentioned in three disorders of Sexual Dysfunction these were related to female specific disorders only, with no references made to partner violence within the male specific sexual disorders. Conduct Disorder and Antisocial Personality Disorders were the only disorders where there was a reference to perpetrating family violence. These were the only times children and spouses were explicitly written about as being relationally impacted by the violent actions of the patient. This is surprising given current good practice requires that all family members, whether victims or perpetrators, and their children, receive timely and appropriate support and treatment to address family violence and its impacts [129].

It needs to be acknowledged that there may be other possible ways of describing family violence which were not elicited by this review. The term “spouse beating”, as already mentioned, emerged only whilst searching for other possible ways of describing family violence. Terms which may be referred to as perpetrating an “assault” or “violent acts/behaviour” towards others were not included in this review. Poor terminology and a lack of referring to family members impacted by an “assault” or “violent acts/behaviour” suggests that this is an area needing urgent attention in future revisions. Similarly, not included enough was the “potential” for perpetrating violence against a family member. This was implied in Alcohol Use Disorder [5] and is interesting to note: “individuals with an alcohol use disorder may continue to consume alcohol despite the knowledge that continued consumption poses significant physical (e.g., blackouts, liver disease), psychological (e.g., depression), social, or interpersonal problems (e.g., violent arguments with spouse while intoxicated, child abuse)” (p. 492).

### 2.2. Referencing Violence within ICD-10

ICD-10 [6] is a manual used worldwide to classify physical and mental diagnoses and to identify factors, such as social circumstances, which may impact them. The World Health Organization states the ICD-10 (in the Purpose and Uses section online) “is the foundation for the identification of health trends and statistics globally, and the international standard for reporting diseases and health conditions. It is the diagnostic classification standard for all clinical and research purposes” [6]. The ICD has been revised several times, with the most recent being the ICD-10 Version: 2016. Used for recording, analysing, interpreting and comparing mortality and morbidity data worldwide, the ICD-10 provides alphanumeric code for disease and other health problems. This system allows for easy storage and retrieval of this data for research purposes [130].

The ICD-10 2016 online edition Section V Mental and behavioural disorders and Section XXI Factors influencing health status and contact with health services was systematically reviewed for any mention of the terms “family violence”, “domestic violence”, “interpersonal violence”, “partner violence”, “violence”, “spouse/spousal beating”, “adverse” and “trauma/tic”. The same method of searching as for the DSM-5 was used for the ICD-10. See DSM-5 section for details. The results can be found in Table 2 below.

The term “family violence” does not appear anywhere in the ICD-10. The term “severe interpersonal violence” is mentioned under Problems in Relationship with Spouse or Partner suggesting that only “severe” interpersonal violence is considered important. This reference to interpersonal violence does not include any mention of infants, children and adolescents who may witness or be impacted by this discord. The term “violence” is mentioned under Problems related to alleged Physical Abuse of Child, however these references are relegated to a section of the ICD-10 which is concerned not with the coding of mental disorder specific criterion, but the description of additional factors which might impact the key mental disorder coding criteria. The term “assault” is used in Section XX—External Causes of Morbidity and Mortality to provide specific examples of injury to another person made and the term “abuse” is mentioned under Maltreatment Syndromes in Section XIX—Injury, Poisoning and Certain Other Consequences of External Causes, in reference to physical, sexual and childhood abuse. Surprisingly, these also do not mention family violence and its role in both abuse and assault. The term “serious mishandling” is mentioned in the criteria for Reactive Attachment Disorder of Childhood, however, this term is not fully explained and no link to family violence is made. When the additional terms of “adverse” and “trauma” were searched, two additional mentions were found. “Adverse experiences” are mentioned in Dissocial Personality Disorder whilst “trauma” is mentioned in the criteria for Post-Traumatic Stress Disorder. However, neither mention “family nor interpersonal violence” as a source of the trauma or adversity experienced.

As with the DSM-5, there are many other diagnostic categories in which family violence could realistically and appropriately have been mentioned, but was excluded from the ICD-10. Within the chapter relating specifically to mental health, Chapter V Mental and Behavioural Disorders, domestic violence was omitted from every diagnosis. This is surprising given the plethora of research suggesting more than a strong correlation between domestic violence and the development of mental health disorders [129,131,132,133,134,135,136,137]. Although not mental health specific, it was also noted that domestic violence could have also been mentioned as a risk factor for complications under Chapter XV Pregnancy, Childbirth and the Puerperium; throughout Chapter XVI—Certain Conditions Originating in the Perinatal Period; Chapter XIX Injury, Poisoning and Certain Other Consequences of External Causes; and Chapter XX External Causes of Morbidity and Mortality. Furthermore, although mentioned twice (see Table 2 for details), domestic violence could have reasonably been mentioned throughout Chapter XXI Factors Influencing Health Status and Contact with Health Services.

### 2.3. Mental Illness in Children and the DC:0-5

Previously known as the DC:0-3 [138], the expanded DC:0-5 was released in 2016 and included a new classification of specific Relationship Disorders in Axis I [7]. This encourages the recognition that difficulties can lay outside of the child, and that relational complexities can occur between one caregiver and the infant/child rather than as a feature of the infant or child’s relationships across all their significant relationships. The DC:0-5 is a 212 page diagnostic classification manual for infants and young children produced by Zero To Three in Washington [7]. The DC:0-5 is a multi-axial system with; Axis I providing clinical diagnoses (grouped into eight categories), Axis II providing the relational context, Axis III providing physical health conditions and considerations, Axis IV psychosocial stressors and Axis V developmental competence. The use of a multi-axial diagnostic system can assist the reader in considering the context around the individual rather than just focusing on the pathology. The DC:0-5 is developmentally sensitive and incorporates the need to assess for risk, as well as protective factors. It is cognisant of the importance of considering the growing infant and young child’s context, culture and relationships in impacting on emerging mental health disorders. Furthermore, as a mental health and developmental disorders manual for early childhood [7], it also readily acknowledges that “behaviours of infants/young children may differ systematically with different caregivers…There are also numerous case reports of symptomatic behaviour in one caregiving relationship that does not generalise to other relationships” (p. 134).

Review of Axis I was conducted using the same systematic method as the DSM-5 and ICD-10. See Table 3 for results.

The inclusion of Relationship Disorders [7] in Axis I and Axis II Relational Context, is intended to encourage practitioners to assess for “cumulative severity of stresses” including “noting their duration and severity” (p. 154). Axis IV Psychosocial Stressors includes the Psychosocial and Environmental Stressor Checklist with 77 different possible stressors listed for clinicians to be aware of when diagnosing mental health disorders. In the introduction section of this Axis domestic violence has been mentioned: “Psychosocial stressors for an infant/young child include acute events and enduring circumstances. Examples of the latter include poverty and domestic violence” [7] (p. 153). In the list “domestic violence” features as one of multiple possible stressors that need to be assessed. Unfortunately, the checklist is designed to measure the number of co-occurring stresses as more “predictive of subsequent maladaptation than any specific stressors” [7] (p. 153). While domestic violence is cited as an example of an enduring psychosocial stressor, its impacts can be understood to be heightened or mitigated by the developmental level of the infant or child, the severity of the violence and the protective buffers offered by the caregivers within that environment “to help the infant/young child understand and cope with the stressor” [7] (p. 153).

As was found with the review of DSM-5 and ICD-10, the DC:0-5 had limited mention of family violence as a risk factor for particular mental disorders. For example, Sensory Over Responsivity Disorder, included “environmental conditions—including lack of movement/tactile stimulation in the early years (e.g., due to being raised in an orphanage, exposure to drugs or prenatal stress, cumulative risk, or community violence—appear to increase risk for Sensory Over-Responsivity Disorder” [7] (p. 441) but made no mention of domestic violence [45,95,110,139]. However, it was specifically mentioned in Anxiety Disorders and Trauma, Stress, and Deprivation Disorders.

## 3. Discussion

This review found that there was a dearth of references to violence occurring within familial relationships across the DSM-5, the ICD-10 and the DC:0-5. By largely omitting an acknowledgement of family violence as a significant risk factor in the development of multiple mental health disorders the enormity of the problem is effectively denied and the opportunity to offer interventions to address the fallout from family violence is missed. Specifically, this ignores the reality that large numbers of adult patients with mental health problems have previously and may also currently be experiencing family violence. Additionally, these same patients may be parents of children who are also exposed to current violence. In particular, the DSM-5 and ICD-10’s failure to recognise the links between early childhood exposure to family violence and mental health disorders, and assessing for current exposure to family violence essentially disappears and therefore disavows the experience of infants, children, adolescents and adults impacted by that violence. By not naming, not seeing and not assessing for family violence, historically or currently, there is no recognition that violence within families impacts all members of that family, no matter their age [140,141]. This averts any need to think about, or take action to intervene now and address the safety needs of the children of the considerable number of adults with mental health disorders who are victims and/or perpetrators of family violence [98,131]. Not considering the children of adult mental health clients who use violence avoids the need to take any action to ensure that those children are safe or to assess their potential need for treatment. Furthermore, the trickledown effect of omitting any significant reference to family violence in adult classification manuals is that within CAMHS, family violence is similarly, often not identified as a serious risk for emerging “mental health problems”.

Terminology within both DSM-5 and the ICD-10 was highly problematic. For example, in searching for terms that may point to the incidence of family violence, words such as “assault”, “serious mishandling”, “adverse events” and “discord” were evident. Their meanings, however, appeared nebulous. The terms were neither fully explained nor the context within which they occurred clearly spelt out. There was also no mention of domestic violence in Section XIX of the ICD-10 which refers to assault as a result of external sources (e.g., drowning, strangulation etc.). Given the reluctance domestic violence victims often have in engaging with services and help seeking and the impacts on physical and mental health [17,118,122,129,142,143,144,145], domestic violence, in its various forms (e.g., intergenerational, spousal, family) should all have separate codes in all three manuals with specific examples of what each form of domestic violence means and be mentioned or referred to in the diagnostic criteria for mental health disorders. Additionally, it would be helpful to have specifier codes which allow clinicians to identify/record if the violence is chronic, acute or improving. Doing so would draw attention to the importance of family and domestic violence in assessing and treating patients and acknowledge the impact on the functioning and engagement of the patient. The World Health Organisation (WHO), which produces the ICD-10 [6], is also the very organisation which has produced repeated reports on the endemic of interpersonal violence across the globe [1,2,3,17]. It would appear judicious for the findings of these WHO produced global reports on interpersonal violence to be meaningfully and responsibly incorporated into the ICD-10, the WHO’s international classification manual “for monitoring of the incidence and prevalence of diseases and other health problems in relation to other variables, such as the characteristics and circumstances of the individuals affected” [130] (p. 3).

It has been noted by Davies [146] that “Medicine’s ambivalence about accepting domestic violence as a key determinant of health is amply highlighted by the absence in our current ICD of any code for domestic violence” (p. 492). That DSM-5 and ICD-10 has not been forthcoming in clearly naming and linking family violence with Mental Disorders perhaps explains, in part, why mental health practitioners also seem reluctant to assess for and acknowledging the impacts of family violence. It can be confronting to think about how to address the legal, statutory, emotional and safety aspects of addressing family violence [147,148]. Yet more challenging in this work is when infants and very young children are involved. Even those working directly in family violence specific services find it difficult to acknowledge and address the impacts of family violence on children in the early years [49,149]. However, CAMHS is in a powerful position to not only champion the importance of the early identification of family violence, particularly in infancy [43,150,151,152,153], but to offer a therapeutic contribution to lives of the families they serve [151,154,155,156,157,158,159,160]. Furthermore, CAMHS has much to offer in how the prevalence and complexities of violence within families is thought about from a mental health perspective [49,161,162,163,164,165].

Not every relational stressor may contribute to mental health difficulties in infancy, childhood, adolescence and beyond. Nevertheless, a stressor as prevalent as family violence is alarming. Part of the role of the CAMHS clinician is to assess for and understand how such stressors impact the subjective and psychological experience of the infant, child or adolescent. Speaking plainly about family violence through clarifying terminology, thinking about, responding to and providing intervention and training programs to address the impacts of family violence, CAMHS, as well as adult mental health may do more than provide early intervention and prevention. This may help make sense of and integrate intersectoral responses where children’s and “women’s experiences of depression, post-traumatic stress, and self-harm can be understood as ‘symptoms’ or the effects of living with violence and abuse. Domestic violence is not just one of many problems, but an issue that requires addressing as a primary concern” [166] (p.223).

### 3.1. Identifying and Responding to Family Violence as a Serious Mental Health Issue

Some twenty years ago calls were made by a group of prominent psychiatrists for the DMS-IV to recognise and create a new classification of relational disorders [167]. They argued that there are some problems that cannot be “understood or described by giving a diagnosis to only one individual” (p. 926) and where violence is present that “violence itself is sufficient to diagnose severe couple dysfunction” [167] (p. 928). Essentially, this group believed that the DSM failed to consider that couples as well as families (which includes children) experience severe dysfunction and believed that “family violence involving child or elder abuse, have been omitted from the DSM-IV” [167] (p. 926). A decade ago the Secretary General of the United Nations commissioned the “World Report on Violence Against Children” [168]. This report unequivocally identified the need for a strategic response to “break down the silence in which most children endure episodes of physical, psychological or sexual violence at home” (p. 81). This was in order to recognise that “children who have experienced family violence have a wide range of treatment needs” [168] (p. 84). More recently there have been calls for the upcoming ICD-11 to screen for intimate violence, although these are limited to partner violence [146,169]. The inclusion of ‘field tested’ diagnostic criteria related to intimate partner violence, Heyman, Slep and Foran [169] argue, could be anticipated to have “even wider and deeper influence in healthcare globally” (p. 78) although they admit “there is no assurance that these, or any other criteria, will be included in ICD-11” (p. 73).

That the DC:0-5 created a new category of Relationship Disorders in 2016 has recognised in part, what the “Committee on the Family” called for in 1995. This is an acknowledgement that “certain problems are relational by their very nature and simply cannot be understood or described by giving a diagnosis to only one individual” [167] (p. 928). This is in contrast to the DSM-5 and ICD-10’s continued implication that psychopathology exists within the individual. The DC:0-5 identifies caregiving relationships as central to understanding the infant and young child. Axis II within the DC:0-5 is embedded within a framework which considers the caregiving and relational context as vital to understand in the assessment of infant and young children’s development and functioning. Despite this, references to domestic violence were concentrated in the Psychosocial and Environmental Stressor Checklist. Again, omission fails to direct clinicians and practitioners to recognise family violence as a serious risk factor, impinging on the development of good mental health for infants, children and adolescents.

The DC:0-5 sits domestic violence within a checklist which measures a number of co-occurring stresses as more “predictive of subsequent maladaptation than any specific stressors” [7] (p. 153). Domestic violence is cited as an example of an enduring psychosocial stressor, its impacts understood to be heightened or mitigated by the developmental level of the infant or child, the severity of the violence and the protective buffers offered by the caregivers within that environment “to help the infant/young child understand and cope with the stressor” [7] (p. 153). Including domestic violence as simply one of multiple, possible stressors to impinge on the young child’s development does not sufficiently alert CAMHS to the complex, often hidden and urgent imperatives which need to be responded to when an infant or young child is living with family violence, nor how to effectively respond to and treat this high-risk issue.

Community, justice based services and community health services have led the way in developing specific treatment responses for children and women impacted by family violence [41,155,170,171,172,173,174,175]. Similarly, community based men’s behaviour treatment programs, developed to address men’s violence, have existed for decades [176,177,178,179,180]. Interventions focusing on reparative work with children and their fathers after family violence is relatively newer territory [158,181] as is any concentrated treatment approach for women who perpetrate violence within intimate relationships, or support programs for men who are victims of family violence [21,22,136,182,183,184]. Considerably fewer treatment programs to address the impacts of family violence have been developed within CAMHS settings, or within mental health generally, but where they have, there is a strong focus on infants [151,156,157,158,162,185,186]. Community based approaches to working with family violence have tended to eschew a recognition of mental health issues or approaches [166,178,187]. Given the high correlation between mental health issues and family violence, there would be much to be gained by bringing these differing services together (including the judicial system) to build new, stronger and more efficacious treatment responses to family violence.

### 3.2. Limitations

It is impossible to capture every conceivable relational stressor that may contribute to mental health difficulties in infancy, childhood, adolescence and beyond. There are other, high prevalence and significant stressors where it could equally be argued greater recognition and acknowledgement is needed within the pages of the DSM-5 and ICD-10. It cannot be definitively argued that family violence has more impact necessarily than another stressor such as early neglect, and/or sexual abuse, both of which carry monumentally damaging risk factors [188,189,190,191,192,193]. The high correlation between one form of adversity and others (for example child abuse, homelessness etc., and family violence) also adds additional complexities not covered in this paper [44,99,194]. The sheer size of these classification manuals, the diversity of issues needing to be covered and the complexities in the uniformity of definitions leaves it open to criticism’s such as have been covered in this paper [9,167,169]. Furthermore, it needs to be noted that the sheer work involved in tabulating the results of the vast number of working groups that contribute to each diagnosis and the lengthy time line between the publication of each version of the DSM hinders its ability to quickly incorporate new research methods such as has “emerged through remarkable advances in new technologies and substantive knowledge in neuroscience” [195] (p. 28).

## 4. Conclusions

The APA declares that DSM-5 (online version) is “the most comprehensive, current, and critical resource for clinical practice available to today’s mental health clinicians and researchers of all orientations. DSM-5 is used by health professionals, social workers, and forensic and legal specialists to diagnose and classify mental disorders” [196]. The ICD-10 claims it “is the foundation for the identification of health trends and statistics globally” [6]. DC:0-5 states that it is not in competition with the ICD-10 and DSM-5 but that “current versions of the latter do not adequately cover syndromes in the earliest postnatal years—syndromes that clinicians encounter and may require urgent attention and preventative interventions” [7] (p. ii). However, all three classification manuals omit clear and consistent references to family violence as a serious risk factor in the development of mental health issues for infants, children, adolescents and adults. Given the extraordinarily high rates of family violence across the world, and the ample evidence of the deleterious impacts of family violence on not just adults, and in particular women [34,118,121,122,132,134,135,197], but on infants, children and adolescents [42,43,45,95,99,100,198,199,200], there is an urgent need for family violence to be appropriately acknowledged and clearly recognised within the pages of DSM and ICD classifications. DSM-5 and ICD-10. These three key international adult and child mental disorder classification manuals are long overdue in needing to acknowledge family violence as a serious risk factor in the development of mental health symptoms and disorders for infants, children and adolescents. Families seeking help from CAMHS cannot wait, however, for such classification manuals to catch up. Infants, children and adolescents and their families affected by family violence need CAMHS to act now.

## Figures and Tables

**Table 1 brainsci-07-00133-t001:** Frequency of the terms “family violence”, “domestic violence”, “interpersonal violence”, “partner violence”, “violence”, “spouse/spousal beating”, “adverse” and “trauma/tic” in the context of family violence and events experienced or perpetrated in the DSM 5 Section II.

Diagnostic Category DSM-5	Number of Times the Following Terms Are Referred to in Text				Direct Quote of the Term from the Text
	“Violence” Experienced ^1^	“Spouse/Spousal Beating”	“Adverse” Events Experienced	“Trauma/Tic” Events Experienced	
Neurodevelopmental Disorders	0	0	0	0	
Schizophrenia Spectrum and Other Psychotic Disorders	0	0	0	0	
Bipolar and Related Disorders	0	0	0	0	
Depressive Disorders	0	0	1	1	Major Depressive Disorder: “Adverse childhood experiences, particularly when there are multiple experiences of diverse types, constitute a set of potent risk factors for major depressive disorder…” p. 166 Premenstrual Dysphoric Disorder. “Environmental factors associated with the expression of premenstrual dysphoric disorder include … history of interpersonal trauma...” p. 173
Anxiety Disorders	0	0	0	0	
Obsessive-Compulsive and Related Disorders	0	0	0	2	Obsessive Compulsive Disorder: “Physical and sexual abuse in childhood and other stressful or traumatic events have been associated with an increased risk for developing OCD” p. 239 Hoarding Disorder: “Individuals with hoarding disorder often retrospectively report stressful and traumatic life events preceding the onset of the disorder or causing an exacerbation” p. 249
Trauma- and Stressor-Related Disorders	6	0	0	DC ^2^	‘Traumatic’ was included repetitively in the diagnostic criteria for Posttraumatic Stress Disorder and Acute Stress Disorder ^3^ Posttraumatic Stress Disorder: (i) “Witnessed events include … domestic violence…” p. 274 (ii) “Children may experience co-occurring traumas (e.g., physical abuse, witnessing domestic violence) and in chronic circumstances may not be able to identify onset of symptomatology” p. 277 (iii) “Peritraumatic Factors … These include … interpersonal violence (particularly trauma perpetrated by a caregiver or involving a witnessed threat to a caregiver in children)” p. 278 (iv) “At least some of the increased risk for PTSD in females appears to be attributable to a greater likelihood of exposure to traumatic events, such as rape, and other forms of interpersonal violence…” p. 278 Acute Stress Disorder: (i) “Witnessed events include… severe domestic violence...” p. 282 (ii) “The increased risk for the disorder in females may be attributable in part to a greater likelihood of exposure to the types of traumatic events with a high conditional risk for acute stress disorder, such as rape and other interpersonal violence” p. 285
Dissociative Disorders	2	0	1	DC	Dissociative Amnesia ^3^: “Dissociative amnesia is more likely to occur with (1) a greater number of adverse childhood experiences, particularly physical and/or sexual abuse; (2) interpersonal violence; (3) increased severity, frequency, and violence of the trauma” p. 300 Depersonalization/Derealisation Disorder: “Other stressors can include … witnessing domestic violence…” p. 304
Somatic Symptom and Related Disorders	1	0	0	2	Introduction to Somatic Symptom and Related Disorders: “A number of factors may contribute to somatic symptom and related disorders. These include … early traumatic experiences (e.g., violence, abuse, deprivation) ...” p. 310 Conversion Disorder: “Onset may be associated with stress or trauma, either psychological or physical in nature.” pp. 319
Feeding and Eating Disorders	0	0	0	0	
Elimination Disorders	0	0	0	0	
Sleep Wake Disorders	0	0	1	2	Nightmare Disorder: (i) “Nightmares occurring after traumatic experiences may replicate the threatening situation (“replicative nightmares”), but most do not” p. 404 (ii) “Individuals who experience nightmares report more frequent past adverse events…but not necessarily trauma” p. 405
Sexual Dysfunctions	DC	0	0	1	Diagnostic Criteria for Female Orgasmic Disorder, Female Sexual Interest/Arousal Disorder and Genito-Pelvic Pain/Penetration Disorder ^3^ “The sexual dysfunction is not better explained by a nonsexual mental disorder or as a consequence of severe relationship distress (e.g., partner violence) …” pp. 429–440 Male Hypoactive Sexual Desire Disorder: “…trauma resulting from early life experiences must be taken into account in explaining the low desire…” p. 442
Gender Dysphoria	0	0	0	0	
Disruptive, Impulse-Control, and Conduct Disorders	1	0	0	1	Intermittent Explosive Disorder: “Individuals with a history of physical and emotional trauma during the first two decades of life are at increased risk…” p. 467 Conduct Disorder: “When individuals with conduct disorder reach adulthood, symptoms of aggression, property destruction, deceitfulness, and rule violation, including violence against co-workers, partners, and children, may be exhibited in the workplace and the home…” p. 473
Substance-Related and Addictive Disorders	0	0	0	2	Inhalant Use Disorder: “Childhood maltreatment or trauma also is associated with youthful progression from inhalant non-use to inhalant use disorder” p. 536 Other (or unknown) Substance Use Disorder: “Risk and prognostic factors for other (or unknown) substance use disorders are thought to be similar to those for most substance use disorders and include… childhood maltreatment or trauma...” p. 580
Neurocognitive Disorders	0	0	0	0	
Personality Disorders	0	1	0	0	Antisocial personality disorder “Individuals with antisocial personality disorder tend to be irritable and aggressive and may repeatedly get into physical fights or commit acts of physical assault (including spouse beating or child beating)” p. 660
Paraphilic Disorders	0	0	0	0	
Other Mental Disorders	0	0	0	0	
Medication-Induced Movement Disorders and Other Adverse Effects of Medication	0	0	0	0	
Other Conditions That May Be a Focus of Clinical Attention ^4^	DC	0	0	1	Adult Maltreatment and Neglect Problems (1) Spouse or Partner Violence, Physical: “This category should be used when non-accidental acts of physical force that result, or have reasonable potential to result, in physical harm to an intimate partner or that evoke significant fear in the partner that have occurred during the past year...” p. 720 (2) Spouse of Partner Violence, Sexual: “This category should be used when forced or coerced sexual acts with an intimate partner have occurred during the past year…” p. 720 Housing Problems, Homelessness: “An individual is considered to be homeless if his or her primary night-time residence is a homeless shelter, a warming shelter, a domestic violence shelter…” p. 723 Other Personal History of Psychological Trauma p. 726
Total ^5^	10	1	3	12	
Total DC	2	0	0	2	

^1^ The following terms were included: “family violence”, “domestic violence”, “interpersonal violence”, “partner violence”, “violence” (when in context of family violence); ^2^ DC: Term included in the Diagnostic Criteria; ^3^ No quote or reduced quotes have been included as there are multiple instances of the term as it is part of the diagnostic criteria; ^4^ Note these are not mental disorders but were included in Section II of the DSM-5; ^5^ DC are not included in the overall frequency total but a frequency of DC is reported in “Total DC”.

**Table 2 brainsci-07-00133-t002:** Frequency of the terms “family violence”, “domestic violence”, “interpersonal violence”, “partner violence”, “violence”, “spouse/spousal beating”, “adverse” and “trauma/tic” in the context of family violence and events experienced or perpetrated in the ICD-10 2016 (online) Edition [6].

Section	Diagnostic Categories	Number of Times the Following Terms Are Referred to in Text				Direct Quote of the Term from the Text
		“Violence” Experienced ^1^	“Spouse/Spousal Beating”	“Adverse” Events Experienced	“Trauma/Tic” Events Experienced	
Section V—Mental and Behavioural disorders						
	Organic, including symptomatic, mental disorders	0	0	0	0	
	Mental and behavioural disorders due to psychoactive substance use	0	0	0	0	
	Schizophrenia	0	0	0	0	
	Schizotypal disorder	0	0	0	0	
	Persistent delusional disorder	0	0	0	0	
	Acute and transient psychotic disorders	0	0	0	0	
	Induced delusional disorder	0	0	0	0	
	Schizoaffective disorders	0	0	0	0	
	other nonorganic psychotic disorders	0	0	0	0	
	Unspecified nonorganic disorders	0	0	0	0	
	Mood (affective) disorders	0	0	0	0	
	Neurotic, stress-related and somatoform disorders	0	0	0	DC ^2^	F43.1 PTSD “…Typical features include episodes of repeated reliving of the trauma in intrusive memories (“flashbacks”), dreams or nightmares, occurring against the persisting background of a sense of “numbness” and emotional blunting, detachment from other people, unresponsiveness to surroundings, anhedonia, and avoidance of activities and situations reminiscent of the trauma … The onset follows the trauma with a latency period that may range from a few weeks to months…”
	Behavioural syndromes associated with physiological disturbances and physical factors	0	0	0	0	
	Disorders of adult personality and behaviour	0	0	0	0	
	Mental retardation	0	0	0	0	
	Disorders of psychological development	0	0	0	0	
	Behavioural and emotional disorders with onset usually occurring in childhood and adolescence	0	0	0	0	
	Unspecified mental disorder	0	0	0	0	
Section XXI actors influencing health status and contact with health services						
	Persons encountering health services for examination and investigation	0	0	0	0	
	Persons with potential health hazards related to communicable diseases	0	0	0	0	
	Persons encountering health services in circumstances related to reproduction	0	0	0	0	
	Persons encountering health services for specific procedures and health care	0	0	0	0	
	Persons with potential health hazards related to socioeconomic and psychosocial circumstances	0	0	0	0	
	Persons encountering health services in other circumstances	0	0	0	0	
	Persons with potential health hazards related to family and personal history and certain conditions influencing health status	DC	0	0	0	Z63.0 Problems in relationship with spouse or partner: “Discord between partners resulting in severe or prolonged loss of control, in generalization of hostile or critical feelings or in a persisting atmosphere of severe interpersonal violence (hitting or striking)”
Total ^3^		0	0	1	0	
Total DC		1	0	0	1	

^1^ The following terms were included: “family violence”, “domestic violence”, “interpersonal violence”, “partner violence”, “violence” (when in context of family violence); ^2^ DC: Term included in the Diagnostic Criteria; ^3^ DC are not included in the overall frequency total but a frequency of DC is reported in “Total DC”.

**Table 3 brainsci-07-00133-t003:** Frequency of the terms “family violence”, “domestic violence”, “interpersonal violence”, “partner violence”, “violence”, “spouse/spousal beating”, “adverse” and “trauma/tic” in the context of family violence and events experienced or perpetrated in the DC:0-5 Axis I.

Diagnostic Category DC:0-5	Number of Times the Following Terms Are Referred to in Text				Direct Quote of the Term from the Text
	“Violence” Experienced ^1^	“Spouse/Spousal Beating”	“Adverse” Events Experienced	‘’Trauma/Tic” Events Experienced	
Neurodevelopmental Disorders	0	0	1	0	“For example, young children raised in adverse caregiving environments, such as institutions or orphanages, have approximately a fourfold risk of ADHD in early childhood compared with non-maltreated pre-schoolers living in families” p. 28
Sensory Processing Disorders	0	0	0	0	
Anxiety Disorders	1	0	0	1	“Risk factors associated with impairing anxiety in early childhood include …environmental factors (e.g., exposure to violence^2^, particularly domestic violence), adverse^2^ life experiences (e.g., medical illnesses requiring hospitalizations and procedures) …” p. 56–57 “mutism that presents suddenly after a major traumatic event should be identified as traumatic mutism, not Selective Mutism” p. 60
Mood Disorders	0	0	0	0	
Obsessive Compulsive and Related Disorders	0	0	0	0	
Sleep, Eating, and Crying Disorders	0	0	0	1	“Compared with adults, nightmares in young children can happen more commonly without an identified traumatic exposure, although the social context is important to consider clinically” p. 96
Trauma, Stress, and Deprivation Disorders	DC ^3,4^	0	0	DC ^4^	“The infant/young child was exposed to significant threat of or actual serious injury, accident, illness, medical trauma, significant loss, disaster, violence (e.g., partner violence, community violence, war or terrorism), or physical/sexual abuse in one or more of the following ways…” p. 115
Relationship Disorders	0	0	0	0	
Total ^5^	2	0	1	2	
Total DC	1	0	0	1	

^1^ The following terms were included: “family violence”, “domestic violence”, “interpersonal violence”, “partner violence”, “violence” (when in context of family violence); ^2^ “violence” and “adverse” were not counted in these instances as “domestic violence” was mentioned separately so these incidences of “exposure to violence” and “adverse life experiences” were not referring to family violence; ^3^ DC: Term included in the Diagnostic Criteria; ^4^ No quote or reduced quotes have been included as there are multiple instances of the term as it is part of the diagnostic criteria; ^5^ DC are not included in the overall frequency total but a frequency of DC is reported in “Total DC”.

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
