# Peer review of "A Diagnosis of Denial: How Mental Health Classification Systems Have Struggled to Recognise Family Violence as a Serious Risk Factor in the Development of Mental Health Issues for Infants, Children, Adolescents and Adults"

_brainsci, 2017, doi:10.3390/brainsci7100133_

Round 1

Reviewer 1 Report

This is, in summary, an interesting review manuscript aimed to investigate how ongoing exposure to family violence significantly enhances the development of psychiatric disorders among children. Specifically, this paper provides a general commentary on the misalignment between current knowledge regarding early brain development and the application of this knowledge in key mental health diagnostic texts in determining, or failing to determine, responses to children impacted by familial violence.

The authors may find my main comments/suggestions as follows.

First, when throughout the Introduction section the authors referred to family violence on infants, children and adolescents, they correctly reported that there are many terms to define this complex phenomenon. However, although the difficulties in universally defining the occurrence of this condition, the family violence against infants, children and adolescents needs to be distinguished in terms of risk and precipitating factors, family background and temperamental/personality traits, given the great heterogeneity of this phenomenon on the different ages. Thus, i suggest to add more information to this regard in order to provide a comprehensive overview of this complex condition for the general readership. In addition, whether children under five are more likely than older children to be exposed to trauma needs to be further clarified. More ahead, the authors referred to altered child developing and insecure attachment; however, more details are requested to this regard.

Importantly, when the authors referred to the science of brain development, they correctly reported that research into the impacts of trauma on the developing brain has been significant in the last years. However, they could also, in my opinion, refer to the relevance of hypothalamic-pituitary-adrenal (HPA) dysfunctions throughout the brain development. Recent evidence documented that HPA axis dysfunctions are involved in the pathophysiology of many diseases, in particular neuropsychiatric conditions. Neuropsychiatric conditions and, in particular, mood disturbances may be associated with various HPA axis activity abnormalities, with important pathophysiological implications. Targeting HPA axis dysfunctions might be a novel strategy to improve the outcomes of these conditions. In order to comprehensively address this topic, i suggest to cite and discuss the recent systematic review/metanalysis of Belvederi Murri and colleagues which was published on Psychoneuro-endocrinology in 2016.

In addition, there are some statements throughout the Discussion section such as: “there is no recognition that violence within families impacts all members of that family, no matter their age” or “there is the need to take action/intervene on the safety needs of the children of adults with mental health disorders who are victims and/or perpetrators of family violence, thus avoiding any action to ensure their safety as well as assess their needs for treatment” that need to be further specified and more adequately supported by adequate references. According to the authors’ expertise, what are the main recipients of violence among the family members? Which type of interventions may be planned to ensure safety and avoid unmet needs in victims and/or perpetrators of family violence?

Moreover, the three Tables reported throughout the main text are useful and enhance the internal coherence of the paper but, unfortunately, they are too long and need to be reduced in length. I suggest to generally summarize their main contents.   

Importantly, the authors should insert the most relevant shortcomings/limitations of the present manuscript as they are completely missing in the current version of the paper. This would guarantee a more critical paper for the general readership. 

Author Response

Dear Reviewer,

Thank you for reviewing our manuscript.  Please find attached our responses.

Kind regards,

Dr Wendy Bunston, Dr Candice Franich-Ray, Ms Sara Tatlow

Reviewer 2 Report

It was a pleasure to read and have a chance to review this Commentary Paper; I hope that its publication provides an avenue to encourage a very important discussion on how we understand, consider, and respond to family members at risk for mental health difficulties related to interpersonal violence. Overall, I felt this was a well-written and thoughtful paper which highlighted critical ways in which we as clinicians and researchers do, and do not, consider context and relationships in mental health diagnosis. I particularly applaud the consideration of the DC:0-5 diagnostic system in this review, as it often does not receive the emphasis that it deserves, even amongst practitioners who work with infants and young children (at least in my country). I appreciated the honest commentary about the challenges in each of the diagnostic systems, particularly related to inconsistent terminology, the emphasis on pathology over causality, and the lack of appreciation that violence within a family can be both a symptom of a mental health problem for one person and a risk factor for individuals within that family – especially for its youngest and its most vulnerable members.

I was surprised that the authors did not choose to make any significant comment on the benefits, potential or otherwise, of the use of a multiaxial diagnostic system in considering relationships and context in individual pathology. Particularly in light of the fact that DSM had comprised a multiaxial system in its previous iteration. A 5-axis system seems to be a particular strength of the DC:0-5, even though domestic violence is not particularly highlighted among the list of psychosocial stressors in Axis IV. This approach seems to, by any means, support the diagnostician to consider risk factors and impacts within the family.

I had a few very minor comments for the writers to consider which I hope might strengthen their work. Specifically:

Line 114 – I’m wondering if you meant the word “warring” as opposed to “waring”

Table 1 – in the section related to “Trauma’ and Stressor-Related Disorders/Post Traumatic Stress Disorder” there is an additional ii) instead of a iii)

Lines 230 to 234 – I appreciate the intent for parallel structure with the repeated use of the word “different”, a word which is also repeated in the following sentence; however, I found the reading of this section to be awkward. Perhaps you could consider an edit.

Lines 443 to 444 – the Psychosocial and Environmental Stressors checklist does appear within the main text related to Axis IV and not in an appendix/addendum.

Line 447 – I believe there may be a word missing (“for” perhaps).

In summary, during my initial read of this commentary, I wondered the authors had expectations that these mental health diagnostic systems (particularly DSM and ICD) should be more than they ever intended to be; however, the thoughtful and persuasive arguments helped me to understand that as clinicians and researchers, we should expect and demand that our diagnostic systems take into account the most recent available evidence about the impact of family violence, not only on the individuals with mental health difficulties, but on the people who share their environments and lives.  I will look forward to seeing this commentary published.

Author Response

Dear Reviewer,

Thank you for reviewing our manuscript.  Please find attached our responses. 

Kind regards,

Dr Wendy Bunston, Dr Candice Franich-Ray and Ms Sara Tatlow
